# Global impact of benthic denitrification on marine $N_2$ fixation and primary production simulated by a variable-stoichiometry Earth system model

Na Li[1], Christopher J. Somes[1], Angela Landolfi[2], Chia-Te Chien[1], Markus Pahlow[1], and Andreas Oschlies[1]

[1]GEOMAR Helmholtz Centre for Ocean Research Kiel, Kiel, Germany
[2]Institute of Marine Sciences (ISMAR), CNR, Rome, Italy

**Correspondence:** Na Li (nli@geomar.de)

**Abstract.** Nitrogen (N) is a crucial limiting nutrient for phytoplankton growth in the ocean. The main source of bioavailable N in the ocean is delivered by $N_2$-fixing diazotrophs in the surface layer. Since field observation of $N_2$ fixation are spatially and temporally sparse, the fundamental processes and mechanisms controlling $N_2$ fixation are not well understood and constrained. Here, we implement benthic denitrification in an Earth System Model of intermediate complexity (UVic-ESCM 2.9) coupled to an optimality-based plankton ecosystem model (OPEM v1.1). Benthic denitrification occurs mostly in coastal upwelling regions and on shallow continental shelves, and is the largest N-loss process in the global ocean. We calibrate our model against three different combinations of observed Chl, $NO_3^-$, $PO_4^{3-}$, $O_2$ and $N^* = NO_3^- - 16PO_4^{3-} + 2.9$. The inclusion of $N^*$ provides a powerful constraint on biogeochemical model behavior. Our new model version including benthic denitrification simulates higher global rates of $N_2$ fixation with a more realistic distribution extending to higher latitudes that are supported by independent estimates based on geochemical data. Oxygen deficient zone volume and water column denitrification rates are reduced in the new version, indicating that including benthic denitrification may improve global biogeochemical models that commonly overestimate anoxic zones. With the improved representation of the ocean N cycle, our new model configuration also yields better global net primary production (NPP) when compared to the independent datasets not included in the calibration. Benthic denitrification plays an important role shaping $N_2$ fixation and NPP throughout the global ocean in our model, and should be considered when evaluating and predicting their response to environmental change.

## 1 Introduction

Nitrogen is a major limiting nutrient for phytoplankton growth throughout the majority of the tropical and subtropical oceans. $N_2$ fixation by photo-autotrophic cyanobacteria supplies the ocean with most of its bio-available N (Fig. 1). The main loss processes of fixed N are denitrification and anammox, which occur under low-oxygen conditions in the water column and sediment pore-waters. In recent decades, numerous studies have been conducted to investigate water-column denitrification and anammox inside oxygen deficient zones (ODZs). These studies have explored various aspects, including genomes, metabolism pathways and their rates, and microbial community structure, utilizing a range of modeling approaches (Hutchins and Capone,

2022). Research on benthic denitrification has been more limited, in spite of the greater part of global ocean denitrification ($\approx 60 - 75\%$) occurring in the sediments (Somes et al., 2013; DeVries et al., 2013; Eugster and Gruber, 2012; Brandes and
25 Devol, 2002). Benthic denitrification is involved in N-cycle stabilizing feedbacks over centennial timescales (Landolfi et al., 2017), and occurs primarily on continental shelves (Middelburg et al., 1996), where a high flux of particulate organic matter fuels the depletion of oxygen in sediment pore-waters. Hence, this process is particularly susceptible to the impacts of human activities. Nevertheless, benthic denitrification is often not evaluated in the analysis of $N_2$ fixation (e.g., Bopp et al., 2022; Hamilton et al., 2020; Paulsen et al., 2017) or not included in current global biogeochemical models (e.g., Pahlow et al., 2020;
Hajima et al., 2020; Dutkiewicz et al., 2015; Landolfi et al., 2013).

In the marine nitrogen cycle, N-loss processes convert bioavailable N to dinitrogen gas that creates a N deficiency compared to phosphate. With their activity $N_2$ fixers replenish the nitrogen deficit in the surface ocean (Somes et al., 2013). Reproducing realistic patterns and rates of denitrification thus will be an important aspect for simulating $N_2$ fixation. In the ocean, a measure of the N deficit is generally expressed by the geochemical tracer N* (Gruber and Sarmiento, 1997), the deviation of nitrate relative to phosphate with respect to the Redfield N:P ratio (Redfield, 1934). Therefore, the distribution of N* has been used
to infer rates of $N_2$ fixation and denitrification (e.g., Gruber and Sarmiento, 1997; Deutsch et al., 2007; Landolfi et al., 2008; Wang et al., 2019). We compare the ability of N* with that of $NO_3^-$ and $PO_4^{3-}$ to constrain parameters of our prognostic model, that includes the representation of large-scale stoichiometric diversity of phytoplankton and diazotrophs (Pahlow et al., 2020).

Projections of how $N_2$ fixation evolves under global warming in current Earth system models are highly uncertain (Wrightson and Tagliabue, 2020). Since $N_2$ fixation can supply the N-limited surface waters with bioavailable nitrogen, it can have a
40 strong impact on how NPP responds to climate change (Bopp et al., 2022; Landolfi et al., 2017). This suggests that a robust understanding of $N_2$ fixation and how it may respond to climate change is essential to predicting future changes in ocean NPP. The global impact of benthic denitrification on $N_2$ fixation and NPP is thus a major focus of this paper.

In this study, we have implemented an empirical parameterization for benthic denitrification (Bohlen et al., 2012) into a
45 global ocean biogeochemical model with an explicit phytoplankton and diazotroph physiology (OPEM, (Pahlow et al., 2020)). We calibrate the model by conducting a large ensemble of simulations, whose parameter sets have been constructed via Latin-hypercube sampling, and select the three best simulations according to an objective cost function based on different combinations of global observational datasets including N*. These simulations significantly improve not only the global N cycle, but also other important aspects of global marine biogeochemistry compared to the original OPEM without benthic denitrification.
We explore model uncertainty by contrasting the behaviour of our three optimized solutions. Finally, we discuss limitations and possible future developments.

## 2 Model description

We use the optimality-based plankton ecosystem model (OPEM, Pahlow et al., 2020) which is incorporated into the UVic model version 2.9 (Weaver et al., 2001; Eby et al., 2009, 2013). Below we provide a brief description of the original OPEM
implementation in UVic 2.9 (Pahlow et al., 2020), followed by the newly-implemented benthic denitrification.

## 2.1 Physical configuration

The physical circulation configuration remains identical to previous UVic 2.9 Kiel versions (Pahlow et al., 2020; Nickelsen et al., 2015). The model has a horizontal resolution of 1.8° latitude × 3.6° longitude, and the ocean component has nineteen levels in the vertical with thicknesses ranging from 50m near the surface to 500m in the deep ocean. The physical circulation model contains the zonally anisotropic isopycnal viscosity and diffusivity schemes to better reproduce equatorial undercurrents (Getzlaff and Dietze, 2013; Somes et al., 2010b). The atmospheric component consists of a simple 2D energy-moisture balance scheme with prescribed wind fields. The physical ocean model is forced by an observation-based monthly wind stress reconstruction (Kalnay et al., 1996). Atmospheric pCO$_2$ is prescribed and kept fixed at 284 ppm.

## 2.2 Optimality-based Plankton Ecosystem Model

The OPEM contains an optimality-based model (Pahlow et al., 2013) for non-N$_2$-fixing phytoplankton and diazotrophs with variable stoichiometry and an optimal current-feeding model (Pahlow and Prowe, 2010) with homeostatic stoichiometry and variable assimilation efficiency for the only zooplankton group. Phytoplankton and facultative diazotrophs allocate their resources optimally to maximize their net growth rates under different ambient environmental conditions. In contrast to the fixed-stoichiometry plankton-ecosystem implementation in UVic 2.9 (Keller et al., 2012), OPEM does not impose a priori a lower potential growth rate on diazotrophs than non-N$_2$-fixing phytoplankton, although the high cost of N$_2$ fixation (Pahlow et al., 2013) has a similar effect. OPEM also accounts for variable stoichiometry of detritus and the associated remineralization. O$_2$ consumption for particulate organic matter decomposition is linked to contributions of C and N remineralization with respiratory quotients of $r_{-O_2:C} = 1.15$ and $r_{-O_2:N} = 2$, respectively.

The temperature dependence is from the OPEM-H configuration of Pahlow et al. (2020), , which applies the Houlton et al. (2008) unimodal temperature function to N$_2$ fixation by diazotrophs. The growth and nutrient uptake of phytoplankton and diazotrophs follow the same exponential temperature function (Eppley, 1972). The temperature dependence in the OPEM configuration is the same as in the original UVic-ESCM, where diazotrophs grow only when temperature is above 15°C. Thus, the only difference between the OPEM and OPEM-H configurations is the temperature dependence of diazotrophs. Half-saturation iron concentrations of phytoplankton and diazotrophs are global constants and calibrated as described in Section 2.4.

## 2.3 Benthic denitrification implementation

We include benthic denitrification as a transfer function, which has been empirically derived from benthic flux measurements (Bohlen et al., 2012). Denitrification scales linearly with the rain rate of particulate organic carbon deposited on the seafloor (RRPOC) and is amplified in environments with low oxygen and high nitrate levels. We apply the transfer function with the parameters obtained by Bohlen et al. (2012), same as those implemented in Somes and Oschlies (2015) but differ slightly from the preliminary implementation in Somes et al. (2013). Anammox is implicitly accounted for in the denitrification estimate since this parameterization is designed to capture total fixed-N loss (Bohlen et al., 2012; Koeve and Kähler, 2010) and our

model does not differentiate between different species of dissolved inorganic nitrogen. We apply a sub-grid scale bathymetry scheme where the effect of the unresolved high-resolution bathymetry is parameterized by multiplying with the area-fraction of the sea-floor in each grid cell (Somes et al., 2013; Somes and Oschlies, 2015). This is crucial for resolving high benthic denitrification rates over continental shelves smaller than the coarse-resolution model bathymetry.

## 2.4 Model calibration

### 2.4.1 Ensemble simulation setup

Our parameter settings are based on the OPEM-H configuration of Pahlow et al. (2020); Chien et al. (2020). In order to allow for a wider range of physiological differences between phytoplankton and diazotrophs, we increase the number of parameters to be calibrated by 5 from the pilot work. Thus, we vary 18 parameters in total (Table 1).

The additional parameters are potential light affinity of phytoplankton ($\alpha_{\text{phy}}$) and diazotrophs ($\alpha_{\text{dia}}$), diazotroph half-saturation constant for Fe ($k_{\text{Fe,dia}}$), diazotroph potential nutrient affinity ($A_{0,\text{dia}}$), and linear increase of sinking speed with depth ($w_{\text{dd}}$). We allow diazotroph and phytoplankton parameters to vary independently, using the same ranges for phytoplankton and diazotrophs and only impose the restriction $1 \leq k_{\text{Fe,dia}}/k_{\text{Fe,phy}} \leq 3$. This allows further decoupling of phytoplankton and diazotroph physiology compared to Chien et al. (2020) and Pahlow et al. (2020), who assumed that $k_{\text{Fe,dia}}$ and $A_{0,\text{dia}}$ co-vary with their phytoplankton equivalents. Additionally, we allow a 50% overlap between the ranges of zooplankton grazing preference for diazotrophs and phytoplankton (Table 1) to enable a certain degree of freedom for these highly-uncertain parameters.

The subsistence nutrient quotas ($Q_{0,\text{ phy}}^{\text{N}}$, $Q_{0,\text{ phy}}^{\text{P}}$, $Q_{0,\text{ dia}}^{\text{N}}$, and $Q_{0,\text{ dia}}^{\text{P}}$) are the minimum quotas for maintaining cellular integrity. The subsistence P quota range is the same for diazotrophs and phytoplankton, while the diazotrophs' subsistence N quota range is much higher than that of phytoplankton, and also than the range used by Pahlow et al. (2020). This setting is more consistent with the earlier estimates (Pahlow et al., 2013) obtained by calibration with lab-culture data for *Trichodesmium* (Holl and Montoya, 2008; Mulholland and Bernhardt, 2005).

We use the ranges of these parameters to generate a Latin-hypercube ensemble of 600 different combinations of the 18 parameters selected for calibration. While the Latin-hypercube method is efficient for evenly sampling a large-dimensional space, our 600 parameter sets provide only a very sparse coverage of this 18-dimensional parameter space, but we use this as a pragmatic choice to obtain information about suitable parameter regions.

Every ensemble member has been spun-up with pre-industrial (AD 1850) boundary conditions with fixed atmospheric pCO$_2$ of 284 ppm for 10,000 years. After the spin-up, we integrate for another year and use annual and monthly means of year 10,001 for model evaluation and analysis. Due to the simple atmospheric model that is driven by prescribed winds, the model has virtually no internal interannual variability, making this short analysis period a pragmatic choice.

### 2.4.2 Cost functions

To assess the model performance with respect to the spatial distributions of dissolved tracers and surface chlorophyll $a$, we apply global misfit metrics $J$ based on a maximum-likelihood estimation (ML) method for parameters, assuming that the errors

for the residuals of log-transformed variables between model simulations and observations follow normal distributions (Chien et al., 2020). Minimizing $J$ ensures the best parameter estimates for the given model configuration.

Instead of calculating residuals between model simulations and observations for each model grid cell, we categorize model simulations and observations into 17 biomes (Fay and McKinley, 2014). Ocean biomes are geographical regions characterized by coherent large-scale patterns in physical and biogeochemical functions (Fay and McKinley, 2014), providing a representation of global ocean biogeography. We represent each variable by two statistical measures per biome: spatial average and variance. Therefore, the residuals comprise the discrepancies in spatial averages and variances for all biomes. In the vertical spatial dimension, we do not make any simplifications and residuals are calculated at each depth layer (k). We calculate the residuals of variables between monthly-averaged simulations and observations to resolve seasonal variations in the upper ocean (0-550m), and between annually-averaged simulations and observations below 550m.

Thus, the calculation of our cost function comprises two components for every depth level in our model,

$$J_k = \boldsymbol{A}_k^T \boldsymbol{\Sigma}_k^{-1} \boldsymbol{A}_k + \boldsymbol{V}_k^T \boldsymbol{Q}_k^{-1} \boldsymbol{V}_k, \qquad k \in \{1, 2, \ldots, 19\} \tag{1}$$

where $\boldsymbol{A}$ is the residual between the spatial means of observations ($\boldsymbol{o}$) and the model ($\boldsymbol{m}$), defined as $\boldsymbol{A}$ = mean($\boldsymbol{o}$) - mean($\boldsymbol{m}$), and $\boldsymbol{V}$ is the residual in the spatial variance (var), defined as $\boldsymbol{V}$ = var($\boldsymbol{o}$) - var($\boldsymbol{m}$). The covariance matrices of $\boldsymbol{A}$ and $\boldsymbol{V}$ are denoted by $\boldsymbol{\Sigma}$ and $\boldsymbol{Q}$, respectively.

We apply the cost function to calibrate model solutions against three different combinations of types of observations. The first is identical to that employed by Chien et al. (2020), who chose four types of observations, i.e., nitrate, phosphate, and oxygen measurements from the World Ocean Atlas 2013 (Garcia et al., 2013a, b) and remote sensing-derived surface chlorophyll concentration (NASA Goddard Space Flight Center et al., 2014). For the second, we use only N* = $NO_3^-$ − 16 · $PO_4^{3-}$ + 2.9 (Gruber and Sarmiento, 1997), which relates to several processes including $N_2$ fixation and denitrification (e.g., Landolfi et al., 2008), to test the efficacy of N* for assessing model solutions. The third is the same as the first, except that we replace nitrate with N*. With these three calibrated solutions, we are able to characterize model uncertainty with respect to which observations are used for calibration.

## 3 Results and discussion

### 3.1 Ranges of global tracers and fluxes of the ensemble simulations

Within the 600 ensemble simulations, many tracer concentrations and fluxes, span wide ranges (Fig. 2) except phosphate because phosphorus is conserved in the model. Globally averaged $NO_3^-$ varies by a factor of 5 and N* varies from -23 to 12.5 mmol m$^{-3}$ for the different parameter combinations. $O_2$ varies by a factor of 2, NPP by a factor of 4, globally integrated $N_2$ fixation rates by a factor of 10, benthic denitrification by a factor of 4, and water-column denitrification from 0 to 276 Tg N yr$^{-1}$. The differences between $N_2$ fixation and denitrification among all simulations range from -11.8 to 5 Tg N yr$^{-1}$, which indicates a slight imbalance at the end of the spin-up. The net fixed-N fluxes relative to $N_2$ fixation rates (N imbalance / $N_2$ fixation) range from -10% to 4% with a median of 0.3%, in line with our steady-state assumption.

The different cost metrics yield similar optimal solutions (lowest cost function) for globally averaged $NO_3^-$, $PO_4^{3-}$, N* and $O_2$ (Fig. 2 and Table 1). The sharp gradients around the optimal solutions of global $NO_3^-$ and N* illustrate the strong constraints the cost function provides for these two tracers. In contrast to the original OPEM-H configuration (Fig. 4 in Chien et al., 2020), the majority of the globally averaged $NO_3^-$ simulations underestimate the observational average. Thus, our new model configuration with benthic denitrification has a stronger tendency to lose fixed N relative to original OPEM-H configuration without benthic denitrification, which one might expect from the incorporation of the additional N loss process.

## 3.2 Best models choice

We identify three optimal model solutions calib_$NO_3^-$, calib_N*_only, calib_N*, based on the three cost functions that include different sets of observations, and compare these with the OPEM-H configuration without benthic denitrification (no_bdeni) (Table 2 and Fig. 2) and observations (Fig. 2).

Our solutions calib_N*_only and calib_N* appear better constrained than calib_$NO_3^-$ (Fig. 2a, b and c). Globally-averaged nitrate concentrations of calib_N*_only and calib_N* are closer to the WOA13 average of 31.0 $\mathrm{mmol\,m^{-3}}$ than calib_$NO_3^-$, although calib_$NO_3^-$ is calibrated directly against observed nitrate, whereas calib_N*_only and calib_N* consider nitrate only indirectly via N*. Globally-averaged oxygen concentrations range from 183.5 to 195.7 $\mathrm{mmol\,m^{-3}}$, slightly above the observed WOA13 value of 176 $\mathrm{mmol\,m^{-3}}$. Interestingly, $O_2$ is also quite well constrained in calib_N*_only (Fig. 2c and Table 2), although it is calibrated only against N*.

While the inventory of nitrogen is essentially identical among the three optimal model solutions, the nitrogen fluxes vary widely. The calibrated estimates for water-column denitrification are more variable (2.8–69.5 $\mathrm{Tg\,N\,yr^{-1}}$) than for benthic denitrification (91.4–105.5 $\mathrm{Tg\,N\,yr^{-1}}$). This relatively weak constraint on water-column denitrification was also reported by Chien et al. (2020), who used an additional objective to constrain water-column denitrification to values above 60 $\mathrm{Tg\,N\,yr^{-1}}$ (no_bdeni in Table 2). Correspondingly, the addition of benthic denitrification allows much higher estimates of global $N_2$ fixation than no_bdeni.

The high variability of water-column denitrification is much reduced in calib_N*_only and calib_N*, ranging from 30.3 to 42.4 $\mathrm{Tg\,N\,yr^{-1}}$, hence varying similarly to our benthic denitrification estimates of 94.4–105.5 $\mathrm{Tg\,N\,yr^{-1}}$. Thus, incorporating N* into the calibration objective helps reduce the uncertainty of ocean nitrogen fluxes, particularly water-column denitrification. Moreover, using N* also yields more reasonable rates of water-column denitrification and global $N_2$ fixation. Globally integrated water-column denitrification has been estimated between 39–77 $\mathrm{Tg\,N\,yr^{-1}}$ (Eugster and Gruber, 2012; DeVries et al., 2012, 2013; Somes et al., 2013; Wang et al., 2019). This suggests that N* provides a better, more independent constraint on water-column denitrification rates, and hence $N_2$ fixation, than simply combining $NO_3^-$ and $PO_4^{3-}$. $NO_3^-$ and $PO_4^{3-}$ are highly correlated, largely following the Redfield N:P ratio. The correlations between different data (Chl, $NO_3^-$, $PO_4^{3-}$ and $O_2$) are already accounted for by our misfit metric (Krishna et al., 2019). However, N* not only accounts for the correlation between $NO_3^-$ and $PO_4^{3-}$, but can be related directly to non-Redfield processes, such as $N_2$ fixation, denitrification, or variations in particulate C:N:P stoichiometry.

With the representation of benthic denitrification in our model and the inclusion of N* into our cost function, global $N_2$ fixation rates are between 136.1–137.7 Tg N $yr^{-1}$, which is close to previous estimates of 137 Tg N $yr^{-1}$ (Deutsch et al., 2007) and 163 Tg N $yr^{-1}$ (Wang et al., 2019). Our estimates also fall within the range of extrapolations of direct measurements, which yield marine $N_2$ fixation rates between 131 and 253 Tg N $yr^{-1}$ (Großkopf et al., 2012; Luo et al., 2012; Landolfi et al., 2018; Shao et al., 2023).

Globally-integrated net primary production (NPP) rates among our three calibrated simulations are consistently lower than that of no_bdeni (88 Pg C $yr^{-1}$), from 52.8 to 63.0 Pg C $yr^{-1}$. This is much closer to observation-based estimates of 52 (satellite-based, Silsbe et al., 2016) and 53 Pg C $yr^{-1}$ (derived from Argo oxygen measurements, Johnson and Bif, 2021). Export production at 130m (model euphotic depth) ranges from 6.9 to 8.1 Pg C $yr^{-1}$, slightly lower than that of no_bdeni (8.7 Pg C $yr^{-1}$, Table 2), yielding higher export efficiencies (export production / NPP) than no_bdeni. Approximate 13% of NPP is exported as sinking particles to the deep ocean in our new solutions (Table 2). Simulated export production at 100m in the Coupled Model Intercomparison Project Phase 5 (CMIP5) and newer CMIP6 models ranges from about 4.5 to 7.5 Pg C $yr^{-1}$ (Bopp et al., 2013; Laufkötter et al., 2016; Fu et al., 2016) and 5 to 12 Pg C $yr^{-1}$ (Séférian et al., 2020; Henson et al., 2022), respectively. Thus, our estimates fall within the CMIP6 spread. Other data-assimilated global ecosystem and biogeochemistry models yield particulate organic carbon export across the 100m depth horizon of 10.64 (Wang et al., 2023), 6.4 (Nowicki et al., 2022), and 6.7 Pg C yr-1 (DeVries and Weber, 2017), yielding export ratios between 12% and 20%.

Considering the constraints of oxygen, NPP (Fig. 2c, g), and particulate N:P (Table 2), our best solution is calib_N*, which is calibrated by the cost function cost_N*.

## 3.3 Global patterns of N fluxes

### 3.3.1 Overview of benthic denitrification

Due to the similarity of global benthic denitrification patterns and rates in our calibrated solutions, we show the mean of benthic denitrification from three calibrated solutions (Fig. 3 and Fig. 4j-l). Benthic denitrification rates are highest in highly productive regions over shallow continental shelves (e.g., South and East China Seas, Bering Sea) (Fig. 3a). These areas experience benthic denitrification rates over 100 times greater than deep-ocean sediments.

The vertical profile of globally integrated benthic denitrification has a sharp increase in the upper ocean with highest rates on shallow continental shelves (< 160m depth) (Fig. 3b), which accounts for approximately 50% of the total benthic denitrification (Table 2). It should be noted that benthic denitrification is not confined to the upper ocean but occurs at all depths. This is in contrast to water-column denitrification that typically occurs in the upper 900m where oxygen deficient zones develop, and $N_2$ fixation confined to the euphotic upper 130m in our model (Fig. A1). The average of calibrated solutions predicts the largest contributions from the sediments in the Pacific and Atlantic Oceans (Fig. 3c).

### 3.3.2 Influence of benthic denitrification on other N fluxes

The global oceanic fixed-N inventory is maintained by balanced supply of N by $N_2$ fixation at the surface ocean and removal by water-column and benthic denitrification. Figure 4 depicts the global N fluxes of our best model solution calib_N* and the flux changes relative to no_bdeni for each of the three solutions. With the implementation of benthic denitrification, global $N_2$ fixation increases, shifting polewards in the Pacific and intensifying in the Atlantic and southern Indian Oceans relative to no_bdeni. The increased $N_2$ fixation thereby compensates for the extra N loss due to benthic denitrification in our newly calibrated simulations. With the outward extension of $N_2$ fixation, however, mild decreases relative to no_bdeni occur in the centers of the subtropical gyres of the Atlantic and Pacific Oceans.

Water-column denitrification declines relative to no_bdeni (Table 2). The two new simulations calib_N*_only and calib_N* show very similar changes, with generally strong decreases in the volume of the eastern tropical North Pacific oxygen deficient zone (ODZ) (Fig. 4g, h). Our model does not reproduce the ODZ in the Indian Ocean, hence the absence of water-column denitrification there. This is also the main reason for the very low rates of $N_2$ fixation predicted for the Northern Indian Ocean, which could be unrealistic. Yet, there is considerable uncertainty about the regional pattern of $N_2$ fixation in the Northern Indian Ocean due to the sparsity of available observations. For example, Shao et al. (2023) found strong $N_2$ fixation rates at only a few of places along the southwest coast of India in the eastern Arabian Sea, Löscher et al. (2020) could find no evidence for $N_2$ fixation in the Bay of Bengal, and vast areas in the Northern and Western Indian ocean remain unsampled. Aligned with the lowest global integrated flux of water-column denitrification of calib_$NO_3^-$, water-column denitrification rates are reduced almost everywhere.

We calculate the basin-scale N fluxes shown in Fig. 5 and Table A1. The net fixed-N flux is negative in the Pacific Ocean and positive in the Atlantic Ocean. Atlantic remains to act as the primary N source for the global ocean, as does no_bdeni. This is mainly caused by relatively strong iron limitation on diazotrophs in the Pacific Ocean that prevents them balancing the high denitrification rates there, which becomes alleviated by in North Atlantic Ocean that receives high atmospheric iron input from Saharan dust (Somes et al., 2010a; Landolfi et al., 2013; Weber and Deutsch, 2014). The very low $N_2$ fixation rates in the South Pacific (Fig. 4a) can be attributed to the underestimated surface dissolved iron concentration in this region. Since our solutions lack water-column denitrification in the Indian Ocean, this area supplies extra nitrogen to the global ocean, even when benthic denitrification is taken into account. In contrast, the Southern Ocean becomes a nitrogen sink when benthic denitrification is included, as $N_2$ fixation does not occur in this area.

### 3.4 Spatial distributions of biogeochemical tracers

### 3.4.1 Vertical nutrient distributions

We now compare the vertical distributions of nutrients and oxygen of our newly calibrated simulations to the WOA13 (Garcia et al., 2013a, b) and no_bdeni, as global averages and for individual ocean basins (Fig. 6). All simulations reproduce the major patterns of the observed climatological vertical distributions of $NO_3^-$ and $PO_4^{3-}$, although N* profiles deviate from the WOA13 average the most. This is due to the smaller range of N* ($-4.5 - 2.5$ mmol/m$^3$; Fig. 6c) compared to those of $NO_3^-$ ($0 - 40$

mmol/m$^3$; Fig. 6a) and $16\times$ PO$_4^{3-}$ ($0 - 48$ mmol/m$^3$; Fig. 6b). The simulation no_bdeni underestimates N* in the upper 1000 meters, whereas the simulations including benthic denitrification and calibrated to N* (i.e. calib_N*_only, calib_N*) better reproduce upper-ocean N* where N cycle fluxes are strongest. In the deep ocean (below 1000m), due to the addition of benthic denitrification, both calib_N*_only and calib_N* depart more from the observation than no_bdeni, especially in the Pacific and Southern Oceans suggesting that benthic denitrification is overestimated there. The greatest deviation of the N* profiles, particularly in the deep ocean, occurs in calib_NO$_3^-$.

Profiles of N* result from the vertical distributions of the processes affecting N*, and the impact of ocean circulation. Denitrification occurs in both the water column and the sediments and imparts a negative signature to N*, whereas N$_2$ fixation imparts a positive signature to N* in the upper ocean. Our calibrated solutions with benthic denitrification yield higher N$_2$ fixation in the euphotic zone and lower water-column denitrification in oxygen-deficient zones (ODZs) compared to no_bdeni, both of which contribute to increasing upper ocean N* compared to no_bdeni, thereby better reproducing observed N* profiles. However, benthic denitrification also occurs at high rates in the upper ocean on the continental shelves, which can compensate the N* effects of N$_2$ fixation and lead to reduced N* relative to no_bdeni in the shallow subsurface ocean (e.g., calib_N*_only in the Atlantic and Indian Oceans). The low vertical N* variability in the Indian Ocean from 200 to 1000m is due to the lack of an ODZ and subsequent water-column denitrification. This is partially counteracted by high benthic denitrification in the upper 200m, causing an upper-ocean N* gradient inconsistent with observations. In order to disentangle the individual effects, we also show the horizontal distribution of N*, which will be addressed in Section 3.4.2.

For O$_2$, all model simulations slightly overestimate global-averaged deep ocean O$_2$, although the vertical patterns are generally consistent with observations. The positive bias in the Southern Ocean (SO) occurs in intermediate layers (300–2500m). Indian Ocean Deep Water (IODW, 1400–3500m) deviations relative to observations has been partly attributed to the overestimation of O$_2$ in SO (Schmidt et al., 2021), as Circumpolar Deep Water (CDW) is the origin of IODW.

Global O$_2$ profiles of our new simulations show increased concentrations relative to no_bdeni in the mesopelagic zone (200–1000m), where the distribution of O$_2$ is largely dominated by the remineralization of particulate organic matter (POM). This is likely caused by slightly decreased global export production described above (Table 2). O$_2$ concentrations in the upper 1000 meters are lower in calib_N* and calib_N*_only than in calib_NO$_3^-$. This may be explained by the higher calibrated remineralization rates ($\nu_{det}$) when the model is constrained with N* (either calib_N* or calib_N*_only) compared to when constrained with NO$_3^-$ (Table 1). Notably, the vertical O$_2$ profiles of calib_N*_only are in close proximity to the observations, implying that N* alone sufficiently constrains the distribution of O$_2$ in our model. As both water-column and benthic denitrification are tightly linked to oxygen, a proper representation of the N* distribution requires a realistic reproduction of the oxygen distribution in the model that resolves major marine biogeochemical processes properly.

### 3.4.2 Lateral distribution of N*

Examining the horizontal distribution of N* at the surface allows us to disentangle the local and regional responses of N* to N$_2$ fixation and benthic denitrification. The inclusion of benthic denitrification considerably improves the reproduction of observed near-surface (50m) N* in the North Pacific Subpolar Gyre (Fig. 7c), which is largely occupied by shallow continental

shelves extending into the Okhotsk and Bering Seas (Fig. 3a). Likewise, near-surface N* in the Arctic ocean is lower than in no_bdeni, more in line with the observations (Fig. 7a-c). Here, however, this improvement is limited, likely due to the coarse resolution. Our model lacks of representation of narrow continental shelves (e.g., among Canadian Arctic archipelago islands) and circulation around them.

The twilight zone collects euphotic zone signals through sinking and remineralization of organic matter. Thus, the non-Redfield stoichiometry of the exported organic matter, local denitrification in both the water column and the sediments, as well as $N_2$ fixation all contribute to the horizontal pattern of N* at 300 m. The low N* signal originating from excessive water-column denitrification at 300m in no_bdeni is mitigated in the the eastern tropical north Pacific ODZ and nearly gone in the the eastern tropical south Pacific ODZ in the new simulations (Fig. 4). The optimal model solutions better simulate elevated N* in the Atlantic Ocean, Indian Ocean, and Western North Pacific (e.g., see calib_N* in Fig. 7f). This is due to the higher contribution of $N_2$ fixation in these regions (Fig. 4) relative to no_bdeni, but also to the elevated euphotic phytoplankton and associated detritus N:P (not shown) with the assumption of identical remineralization rates of detrital N and P. Compared to the observations, our model underestimates N* in the western and central parts of the South Pacific subtropical gyre. This signal is driven both by underestimated $N_2$ fixation rates (Fig. 4a) and low N:P ratios of phytoplankton and corresponding detritus N:P (<13.5) compared to the Redfield ratio (molar N:P = 16) that is applied in the calculation of N*.

The influence of local biogeochemical processes, such as benthic denitrification, on N* is scarcely discernible at 2000m. The distribution of N* in the deep ocean reflects global biogeochemical signals accumulating over decades to millennia along the thermohaline circulation (Fripiat et al., 2021; DeVries and Primeau, 2011), thus diluting the local flux signals. Our simulations constrained by N* (calib_N*_only and calib_N*) reproduce the basin gradient visible in the WOA 2013 data best (see calib_N* in Fig. 7h).

### 3.4.3 Patterns of $O_2$

In the core of the ODZs ($\approx$300m), the $O_2$ distribution varies similarly across all calibrated new simulations compared to no_bdeni (not shown). Low- and mid-latitude oceans have higher subsurface oxygen concentrations (Fig. 8), resulting in less intense ODZs and water-column denitrification (Fig. 4). The simulated and observed spatial patterns are broadly comparable, with the exception of the Arabian Sea, where the observations reveal the presence of a perennial ODZ (Fig. 8a). The lower ODZ volume when including benthic denitrification (Table 2) implies that including benthic denitrification may improve the representation of ODZs in global ocean biogeochemical models that typically overestimate their volume (Cabré et al., 2015).

The overestimation of $O_2$ in calib_N* compared to WOA13 at 2000m occurs mainly in the Pacific Ocean and the change relative to no_bdeni at this depth is spatially homogeneous. While our model solutions show consistent changes in $O_2$ concentration relative to no_bdeni that broadly point towards better agreement with observations at 300m, changes often are in a direction diverging from observations at 2000m (not shown). This finding is also reflected in the vertical profiles of globally averaged $O_2$ shown above (Fig. 6).

## 3.5 Global patterns of C fluxes

Our definition of global NPP includes the sum of phytoplankton NPP and diazotroph NPP. Export production (EP) is calculated as the product of the biomass of sinking particles and their sinking speed at 130m depth. These sinking particles consist of detritus derived from phytoplankton, diazotrophs, and zooplankton. High production occurs in equatorial and subpolar regions (Fig. 9a, e), with the highest NPP found in the equatorial oceans. In contrast, EP in equatorial and subpolar oceans are comparable. The global distribution of export efficiency (EP/NPP, hereafter e-ratio) exhibits a negative relationship between NPP and e-ratio. Low e-ratios (<0.2) occur in low latitudes and high e-ratios (>0.2) in high latitudes, with a few exceptions in the Atlantic and Pacific subtropical gyres (Fig. 9i). Its general pattern is similar to the observational estimate of Dunne et al. (2005) and the model estimate of Henson et al. (2015). The low e-ratios of equatorial oceans result from elevated particle decomposition rates in high-temperature environments, owing to both increasing zooplankton respiration and detritus remineralization with temperature in our model.

Compared to no_bdeni, the new simulations exhibit notable decreases of NPP, mainly located in and adjacent to these high-production regions (Fig. 9b-d). While the primary pattern observed in the distribution of EP is a decline in most areas, there are some small areas that contain an increase (Fig. 9f-h). The decline in EP results in elevated concentrations of oxygen in the underlying subsurface ocean (Fig. 8c). Among the three solutions, calib_N* and calib_N*_only exhibit more similar patterns with each other than with calib_NO$_3^-$.

## 4 Model and calibration limitations

In the northern Indian Ocean, our model shows a too low N* signal due to benthic denitrification at the Bay of Bengal, whereas the WOA13 data indicates an intense ODZ and robust water-column denitrification in the Arabian Sea (Fig. 7d). This is likely due to underestimated coastal upwelling by the coarse model resolution in the Arabian Sea. An examination of CMIP5 models also reveals the presence of systematic deficiencies in the oceanic physics of these Earth System Models (ESMs), resulting in an inaccurate representation of the east-west O$_2$ gradient in the northern Indian Ocean (**?**). The absence of a persistent ODZ in the Arabian Sea in our model is likely a reason why our global water-column denitrification and N$_2$ fixation rates are on the low-end of observational estimates, between 131 and 253 Tg N yr$^{-1}$ (Großkopf et al., 2012; Luo et al., 2012; Landolfi et al., 2018; Shao et al., 2023).

The absence of an Arabian Sea ODZ may also be affected by the biome resolution of our calibration method, which has only one biome for the entire Indian Ocean, filtering out sub-basin variability. Sub-dividing the Indian Ocean biome might have the potential to improve the ability of the cost function to constrain O$_2$ and water-column denitrification. Future research building on high-resolution earth system models may be amenable to choose a spatially finer calibration in terms of 56 biogeochemical provinces (Longhurst, 2007). Concerning the unique characteristic of O$_2$ among tracers, an additional improvement would be to assume a normal distribution of O$_2$ concentration in the global ocean, which represents the data in WOA13 better than the log-normal distribution used here. However, since calib_N*_only chosen by cost_N* without calibrating against O$_2$ fails to reproduce the intense water-column denitrification in the North Indian Ocean as well, further investigation is required.

The vertical distribution of N* shows that simulated N* tends to underestimate the observed N* at depth, particularly below 1000m (Fig. 6c). There are some promising developments that could be implemented to improve the deep-ocean N* distribution. Our cost function metrics tend to focus on the upper ocean by including seasonal-scale variability only for the upper 550m. This artificial emphasis on the upper ocean would be balanced by a refined volume-weighted cost function. In the current model configuration, the sinking velocity increases linearly with depth all the way to the ocean floor, which may lead to an overestimation of benthic denitrification in the deep ocean. A linear increase only up to 1000 meters (Lam and Marchal, 2015) could give us a better representation of global carbon fluxes and corresponding benthic denitrification rates, which is the only plausible nitrogen loss flux that could occur below 1000m (Fig. A1). Likewise, incorporating dissolved organic matter dynamics or preferential remineralization of phosphorus could assist in the reproduction of vertical gradients of N* (Somes and Oschlies, 2015). Improvements to the upper ocean may also have the potential to improve the deep-ocean performance, such as including the sinking speed of particles at the ocean surface ($w_{d0}$) in addtion to its increase with depth ($w_{dd}$) into the calibrated parameters. By taking both $w_{d0}$ and $w_{dd}$ into account for the calibration, the particle flux profile in our model could possibly be represented more accurately.

The influence of particle fluxes on N* via the effect on benthic denitrification leads to a further discussion of the model uncertainties in the representation of benthic denitrification. A first uncertainty in the model estimate of benthic denitrification via the parameterization employed results from the uncertainties in the simulated bottom-water $O_2$ and $NO_3^-$ concentrations and the organic carbon rain rate. The accurate simulation of the rain rate is one of the most critical issues in ocean biogeochemistry and is associated with a high uncertainty (Clements et al., 2022; Kiko et al., 2017). A promising option could be to resolve the dependency of remineralization rate on $O_2$, which could contribute to an accurate representation of the unique vertical profiles observed in ODZs (Pavia et al., 2019; Engel et al., 2022). Such efforts are necessary for a proper investigation of the relation between benthic denitrification and water-column denitrification. Moreover, the empirically-derived parameterization of benthic denitrification itself is subject to a certain uncertainty, albeit to a lesser extent (Bohlen et al., 2012). Ocean physics introduces additional uncertainty. For example, the low physical resolution of the existing UVic model framework imposes relatively little computational demand, but the representation of N and C fluxes could benefit from a higher spatial resolution.

## 5 Conclusion

In order to explore the sensitivity to prior assumptions about which data are used for calibration, we applied our cost function to calibrate model solutions against three different combinations of observations. The best model solution calib_N* reaches the lowest cost function with input of Chl, $PO_4^{3-}$, $O_2$ and N* $= NO_3^- - 16PO_4^{3-} + 2.9$. Model solution calib_N*_only shows significant parallels with calib_N* in various biogeochemical fluxes and tracers. Compared with the canonical cost function cost_$NO_3^-$, that calibrates the model against Chl, $PO_4^{3-}$, $O_2$ and $NO_3^-$, as was done previously (Pahlow et al., 2020; Chien et al., 2020), including N* provide better constraints on globally averaged N* and nitrate, and also on globally integrated water-column denitrification and thus $N_2$ fixation. The greater constraining capacity of N* in comparison to considering nitrate and phosphate separately highlights the importance of accounting for correlations among variables within the cost function (Krishna

et al., 2019) and demonstrates the power of diagnostic tracers such as N* for diagnostic studies of the ocean nitrogen cycle (DeVries et al., 2013; Eugster and Gruber, 2012; Deutsch et al., 2007).

UVic-OPEM achieves a better representation of $N_2$ fixation and N* by incorporating benthic denitrification. Our model configurations demonstrate higher estimates of global $N_2$ fixation and an extension of $N_2$ fixation to higher latitudes in the Pacific, Atlantic and Indian Oceans compared to the simulation excluding benthic denitrification (no_bdeni). The calibrated
model solutions calib_N* and calib_N*_only yield a global $N_2$ fixation of 137.7 and 136.1 Tg N $yr^{-1}$ respectively. The most apparent improvements of modelled N* distribution compared to WOA13 (Garcia et al., 2013a) are located in the surface layers of the North Pacific subpolar gyre, where consistently low N* signals result from the newly-added benthic denitrification in the absence of local changes from other N fluxes and transfers. The improved representation of the global nitrogen cycle enables a more precise reproduction of net primary productivity. The estimated global integrated NPP, ranging from 52.8 to
63.0 Pg C $yr^{-1}$, is consistent with the estimates derived from satellite and Argo-float observations (Silsbe et al., 2016; Johnson and Bif, 2021). The most significant decrease in NPP occurs in the tropical oceans, with a concomitant contraction of oxygen-deficient zones (ODZs). Benthic denitrification plays a globally important role shaping $N_2$ fixation and NPP throughout the global ocean and should be considered in marine biogeochemical models when trying to understand and predict changes in $N_2$ fixation and marine C fluxes.

**Table 1.** Parameter names, ranges, units and descriptions.

| Symbol | Range | Reference range | calib_NO$_3^-$ | calib_N*_only | calib_N* | Reference (no_bdeni) | Definition and units |
|---|---|---|---|---|---|---|---|
| $A_{0,\text{phy}}$ | 200–400 | 120–280 | 365 | 339 | 205 | 229 | phytoplankton pot. nutr. affinity (m$^3$ molC$^{-1}$d$^{-1}$) |
| $A_{0,\text{dia}}$ | 200–400 | — | 311 | 383 | 321 | 171 | diazotroph pot. nutrient affinity (m$^3$ molC$^{-1}$d$^{-1}$) |
| | | | | $(= 0.75 \times A_{0,\text{phy}})$ | | | |
| $\alpha_{\text{phy}}$ | 0.4–0.6 | — | 0.52 | 0.40 | 0.40 | 0.4 | phytoplankton pot. light affinity (m$^2$W$^{-1}$molC gChl$^{-1}$d$^{-1}$) |
| $\alpha_{\text{dia}}$ | 0.4–0.6 | — | 0.51 | 0.47 | 0.58 | 0.5 | diazotroph potential light affinity (m$^2$W$^{-1}$molC gChl$^{-1}$d$^{-1}$) |
| $Q_{0,\text{phy}}^N$ | 0.04–0.06 | 0.04–0.06 | 0.05282 | 0.05396 | 0.05879 | 0.04128 | phytoplankton subsistence N quota (mol molC$^{-1}$) |
| $Q_{0,\text{dia}}^N$ | 0.10–0.16 | 0.06–0.12 | 0.1089 | 0.1246 | 0.1231 | 0.067 | diazotroph subsistence N quota (mol molC$^{-1}$) |
| $Q_{0,\text{phy}}^P$ | 2–4 | 1.3–2.3 | 2.1 | 2.2 | 2.6 | 2.2 | phytoplankton subsistence P quota (mmol molC$^{-1}$) |
| $Q_{0,\text{dia}}^P$ | 2–4 | 2.5–3.5 | 2.56 | 2.08 | 3.58 | 2.71 | diazotroph subsistence P quota (mmol molC$^{-1}$) |
| $k_{\text{Fe,phy}}$ | 0.04–0.08 | 0.04–0.08 | 0.079 | 0.042 | 0.040 | 0.066 | phytoplankton half-satur. const. for Fe (µmol m$^{-3}$) |
| $k_{\text{Fe,dia}}$ | 0.08–0.12 | — | 0.093 | 0.118 | 0.119 | 0.132 | diazotroph half-satur. const. for Fe (µmol m$^{-3}$) |
| | | | | $(= 2 \times k_{\text{Fe,phy}})$ | | | |
| $g_{\text{max}}$ | 1–2 | 1–2 | 1.82 | 1.47 | 1.26 | 1.75 | zooplankton max. specific ingestion rate (d$^{-1}$) |
| $\phi_{\text{phy}}$ | 100–200 | 100–200 | 178 | 146 | 145 | 118 | capture coefficient of phytoplankton (m$^3$ molC$^{-1}$) |
| $\phi_{\text{dia}}$ | 150–250 | 150–250 | 215 | 202 | 232 | 232 | capture coefficient of diazotroph (m$^3$ molC$^{-1}$) |
| $\phi_{\text{det}}$ | 20–100 | 20–100 | 43 | 72 | 92 | 94 | capture coefficient of detritus (m$^3$ molC$^{-1}$) |
| $\phi_{\text{zoo}}$ | 100–200 | 100–200 | 118 | 149 | 156 | 118 | capture coefficient of zooplankton (m$^3$ molC$^{-1}$) |
| $\lambda_{0,\text{phy}}$ | 0.01–0.03 | 0.01–0.03 | 0.016 | 0.016 | 0.014 | 0.018 | T-dependent leakage of phytoplankton (d$^{-1}$) $= M_{0,\text{dia}}$: diazotroph T-dependent mortality rate |
| $w_{\text{dd}}$ | 0.03–0.06 | — | 0.0387 | 0.0391 | 0.0568 | 0.06 | linear increase of sinking speed with depth (m$^{-1}$) |
| $\nu_{\text{det}}$ | 0.04–0.09 | 0.04–0.09 | 0.058 | 0.073 | 0.080 | 0.087 | remineralisation rate at 0°C (d$^{-1}$) |
| $M_{0,\text{phy}}$ | — | — | 0.03 | 0.03 | 0.03 | 0.03 | phytoplankton T-independent mortality rate (d$^{-1}$) |
| $\lambda_{0,\text{dia}}$ | — | — | 0 | 0 | 0 | 0 | diazotroph T-independent leakage rate (d$^{-1}$) |

The reference range refers to the calibration range in Pahlow et al. (2020); Chien et al. (2020).

no_bdeni refers to OPEM_H in the previous studies (Pahlow et al., 2020; Chien et al., 2020).

**Table 2.** Tracers, fluxes, and costs of OPEM simulations.

| variable | model configuration | | | | units | input variables |
|---|---|---|---|---|---|---|
| | no_bdeni | calib_$NO_3^-$ | calib_N*_only | calib_N* | | |
| average $NO_3^-$ | 31.3 | 29.3 | 30.5 | 30.4 | $mmol\,m^{-3}$ | |
| average $O_2$ | 188 | 195.7 | 194.2 | 183.5 | $mmol\,m^{-3}$ | |
| average N* | −1.17 | −2.55 | −1.27 | −1.44 | $mmol\,m^{-3}$ | |
| average particulate N:P | 15.49 | 16.15 | 16.42 | 16.15 | $mol\,mol^{-1}$ | |
| ODZ[†] volume | 174 | 0.3 | 7.9 | 6.6 | $\times 10^{14} m^3$ | |
| $N_2$ fixation | 69.5 | 95.5 | 136.1 | 137.7 | $Tg\,N\,yr^{-1}$ | |
| Water-column Deni. (W) | 69.5 | 2.8 | 42.4 | 30.3 | $Tg\,N\,yr^{-1}$ | |
| Benthic Deni. (B) | 0 | 91.4 | 94.4 | 105.5 | $Tg\,N\,yr^{-1}$ | |
| B/W | – | 32.6 | 2.2 | 3.5 | 1 | |
| B$_{continental\_shelf}$/B | – | 0.5 | 0.6 | 0.5 | 1 | |
| NPP | 88 | 52.8 | 63.0 | 59.3 | $Pg\,C\,yr^{-1}$ | |
| Export production (at 130m) | 8.7 | 6.9 | 8.1 | 7.9 | $Pg\,C\,yr^{-1}$ | |
| EP/NPP | 9.9 | 13.1 | 12.9 | 13.3 | % | |
| cost_$NO_3^-$ | 187.2 | 165 | 202 | 222 | $\times 10^5$ | $NO_3^-$, $PO_4^{3-}$, $O_2$, Chl |
| rank | 4[‡] | **1** | 20 | 30 | | |
| cost_N*_only | – | 37.1 | 16.7 | 18.1 | $\times 10^5$ | N* |
| rank | – | 32 | **1** | 3 | | |
| cost_N* | – | 118.9 | 79.1 | 77.3 | $\times 10^5$ | N*, $PO_4^{3-}$, $O_2$, Chl |
| rank | – | 41 | 3 | **1** | | |

[†] We define ODZ as the region with $O_2$ concentration < 5 $mmol\,m^{-3}$ and here we present the annual average of ODZ volume for each solution.

[‡] no_bdeni is the 4th best solution among the 400 simulations conducted without benthic denitrification in the previous studies (Pahlow et al., 2020; Chien et al., 2020). All other rank numbers refer to the positions among the newly generated 600 simulations that incorporate benthic denitrification. cost_$NO_3^-$ in this study is the same as the cost function in the previous studies.

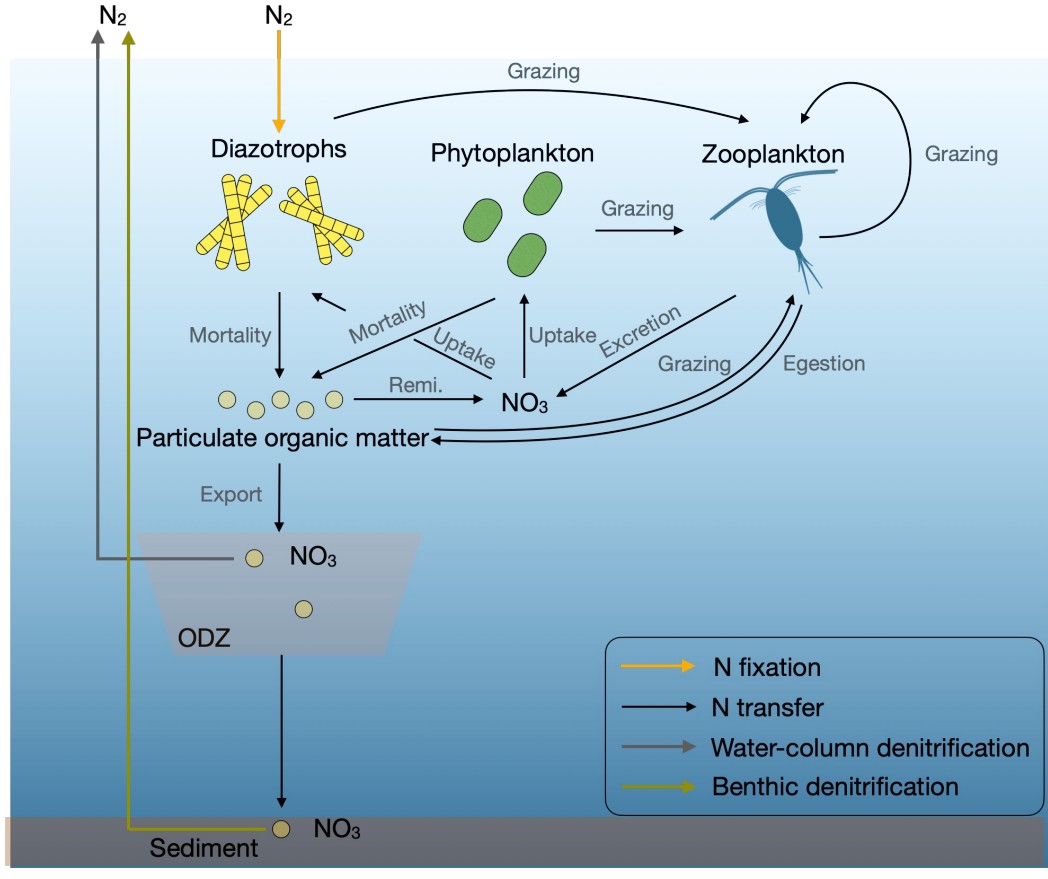

**Figure 1.** Marine N flows in our model. Remi. = remineralization. ODZ = oxygen deficient zone.

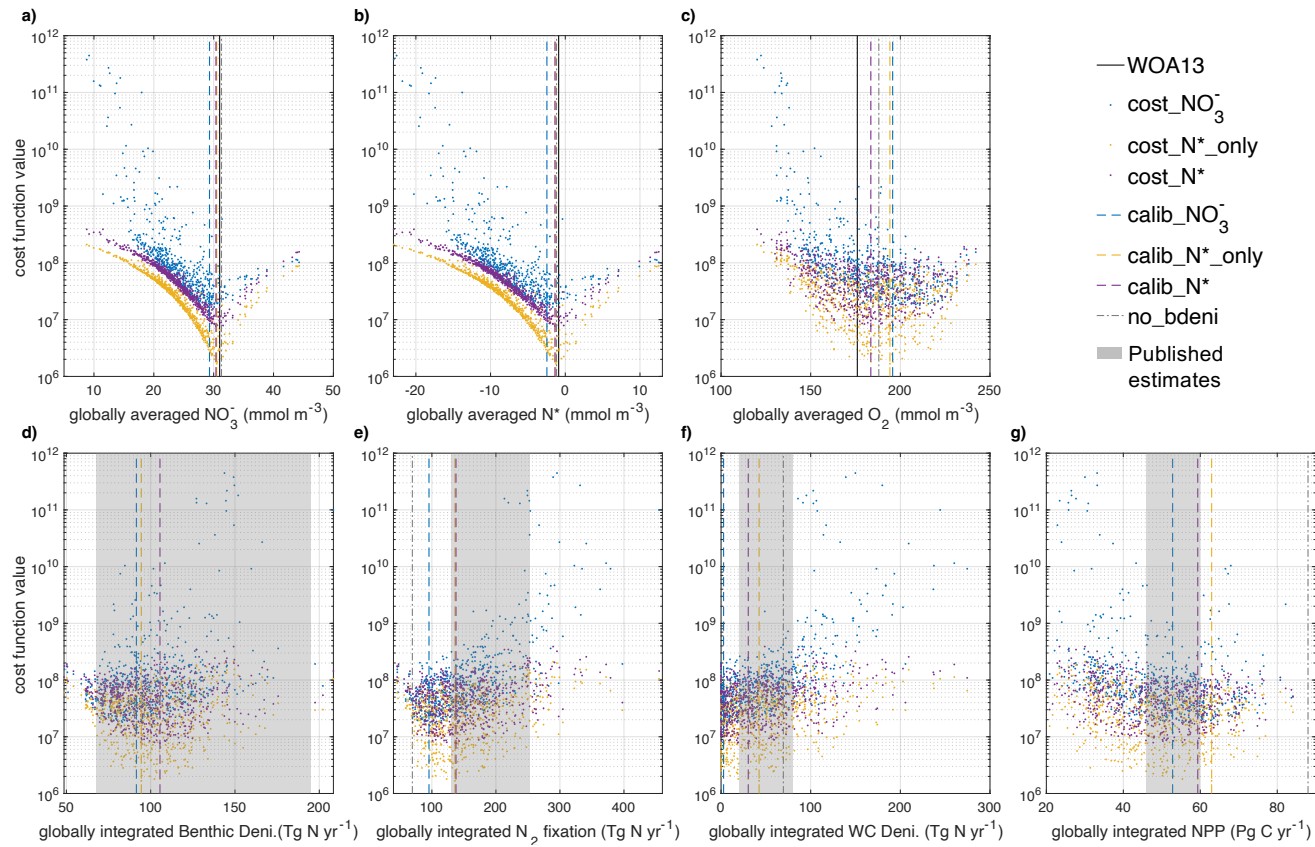

**Figure 2.** Relations between global averages of tracers, integral of fluxes and cost function values. The x-axis of each panel displays the ranges of (**a**) $NO_3^-$, (**b**) N*, (**c**) $O_2$ and (**d**) benthic denitrification, (**e**) $N_2$ fixation, (**f**) water-column denitrification, (**g**) NPP of our ensemble simulations. The calibrated solutions are displayed as dashed lines with colors that correspond to their respective cost functions (cost_$NO_3^-$, cost_N*_only, cost_N*). In order to compare, we also depict the values of tracers from WOA13 data (Garcia et al., 2013a, b) with solid black lines, whereas the tracers and fluxes from the simulation no_bdeni with dash-dotted grey lines. Deni. is short for denitrification and WC. for water-column. Published estimates of global benthic denitrification, $N_2$ fixation, water-column denitrification and NPP range from 68 to 195 Tg N $yr^{-1}$ (DeVries et al., 2013; Bohlen et al., 2012; Eugster and Gruber, 2012; Wang et al., 2019), from 131 to 253 Tg N $yr^{-1}$ (Großkopf et al., 2012; Luo et al., 2012; Landolfi et al., 2018; Shao et al., 2023), from 20 to 80 Tg N $yr^{-1}$ (Wang et al., 2019; DeVries et al., 2013; Somes et al., 2013; Bianchi et al., 2012; Eugster and Gruber, 2012) and from 46 to 60 Pg C $yr^{-1}$ (Johnson and Bif, 2021; Silsbe et al., 2016), respectively.

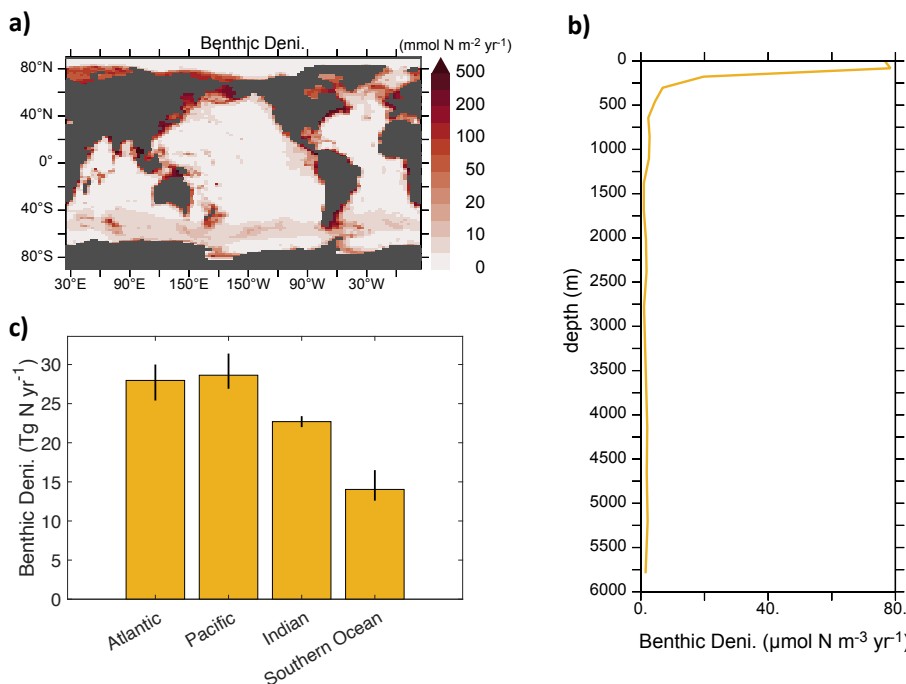

**Figure 3.** Spatial distributions of benthic denitrification, based on the mean values obtained from our calibrated solutions. (**a**) Geographic distribution of the vertical integrals. (**b**) Vertical profile of the global averages. (**c**) Four ocean basin integrals. Southern Ocean is defined as the body of water located to the south of the 40°S latitude. The error bar in panel (**c**) indicates the uncertainty window among the calibrated solutions (i.e., calib_$NO_3^-$, calib_N*_only and calib_N*).

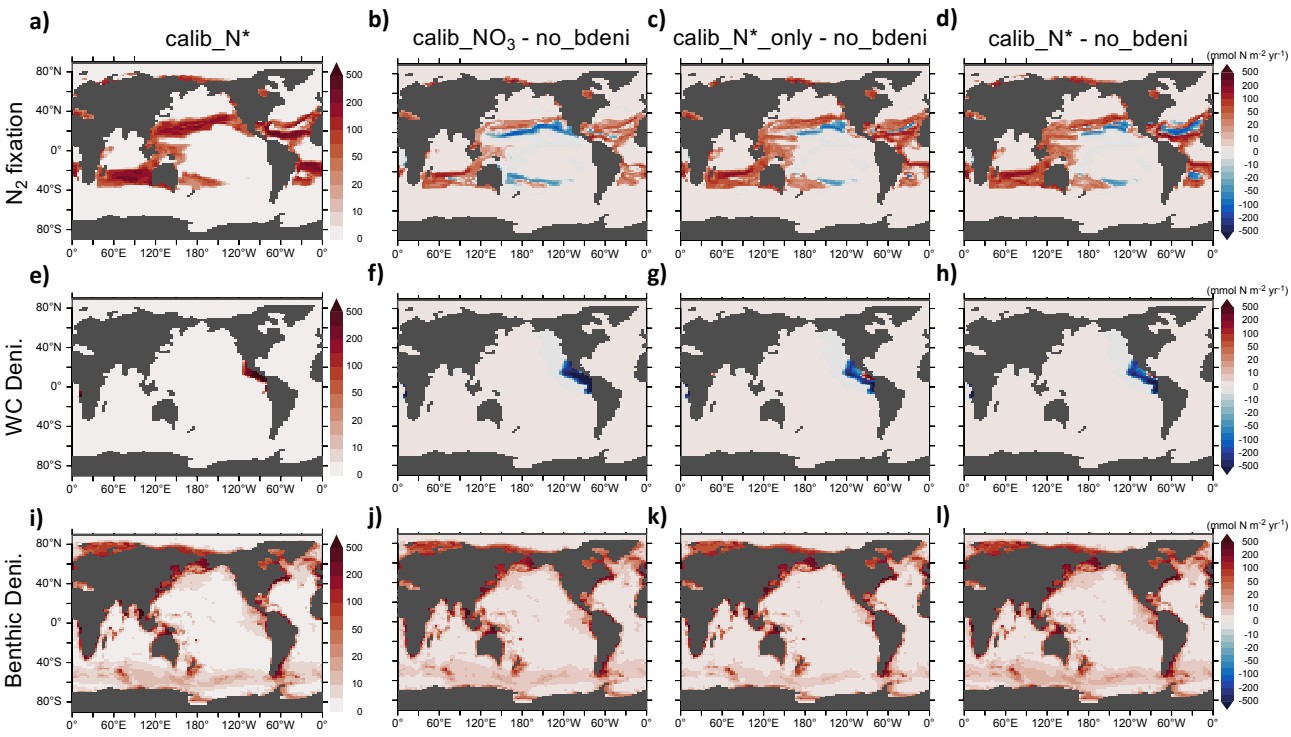

**Figure 4.** $N_2$ fixation, water-column denitrification and benthic denitrification distributions of calib_N* and the changes of three new solutions relative to no_bdeni.

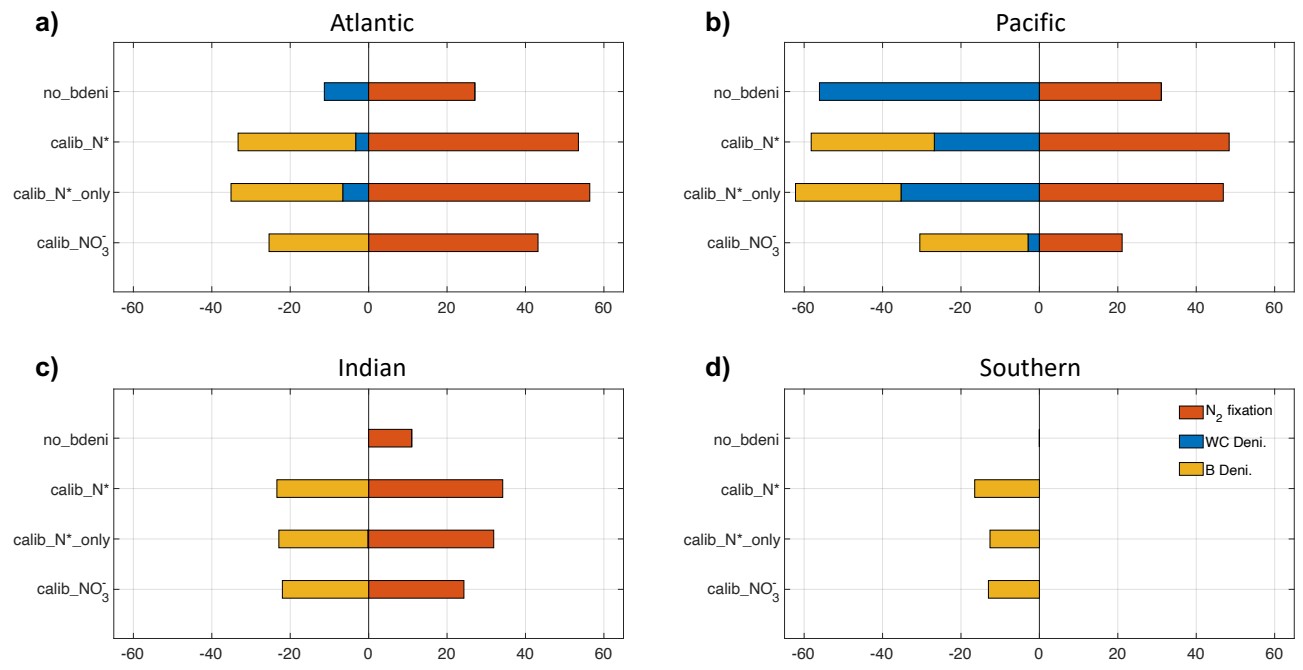

**Figure 5.** Basin-scale bio-available N fluxes in the Atlantic (a), Pacific (b), Indian (c) and Southern (d) oceans, including $N_2$ fixation, water-column denitrification and benthic denitrification.

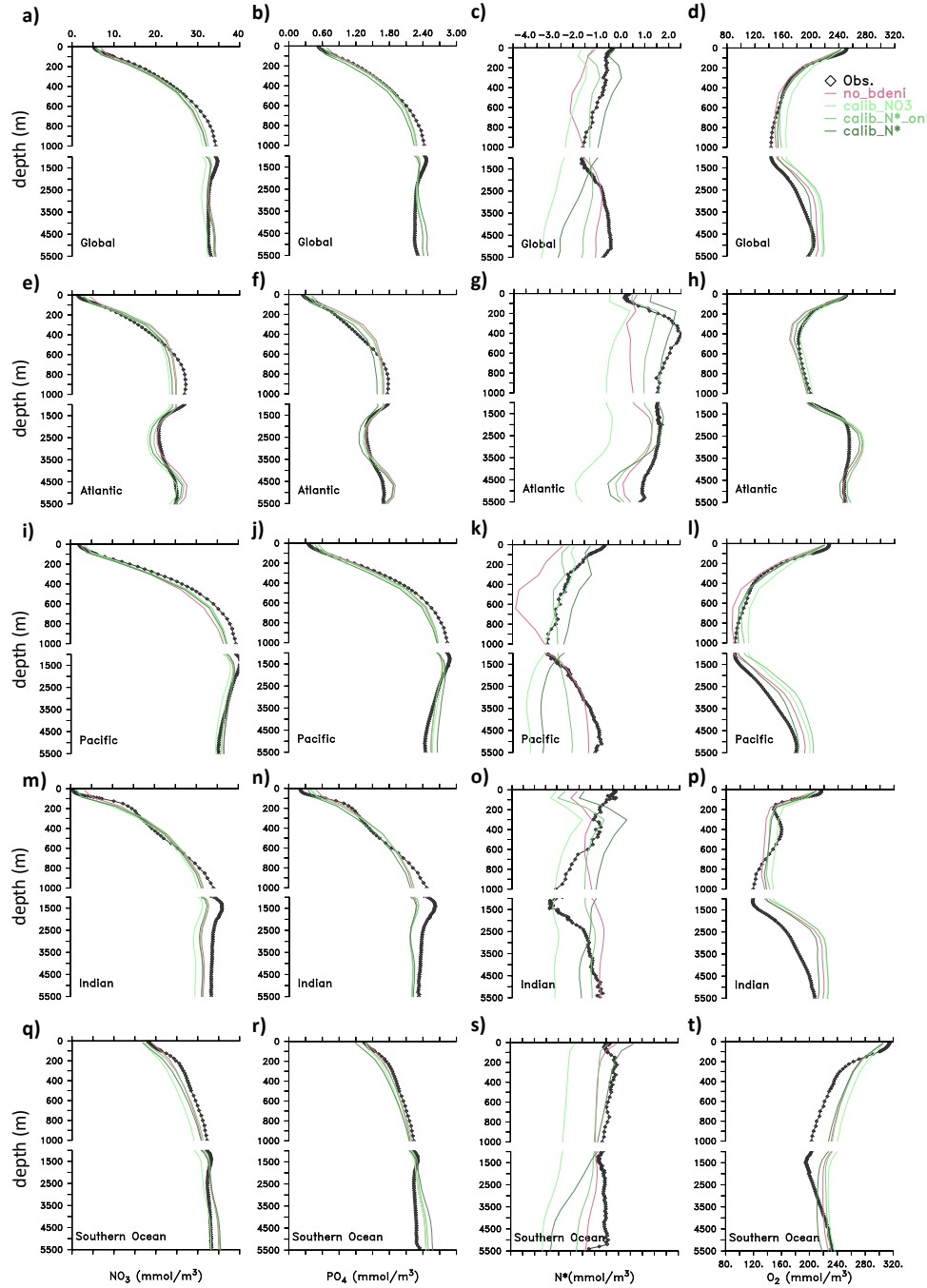

**Figure 6.** Vertical distributions of tracers ($NO_3^-$, $PO_4^{3-}$, N*, $O_2$) in the global ocean and the Atlantic, Pacific, Indian, Southern Oceans.

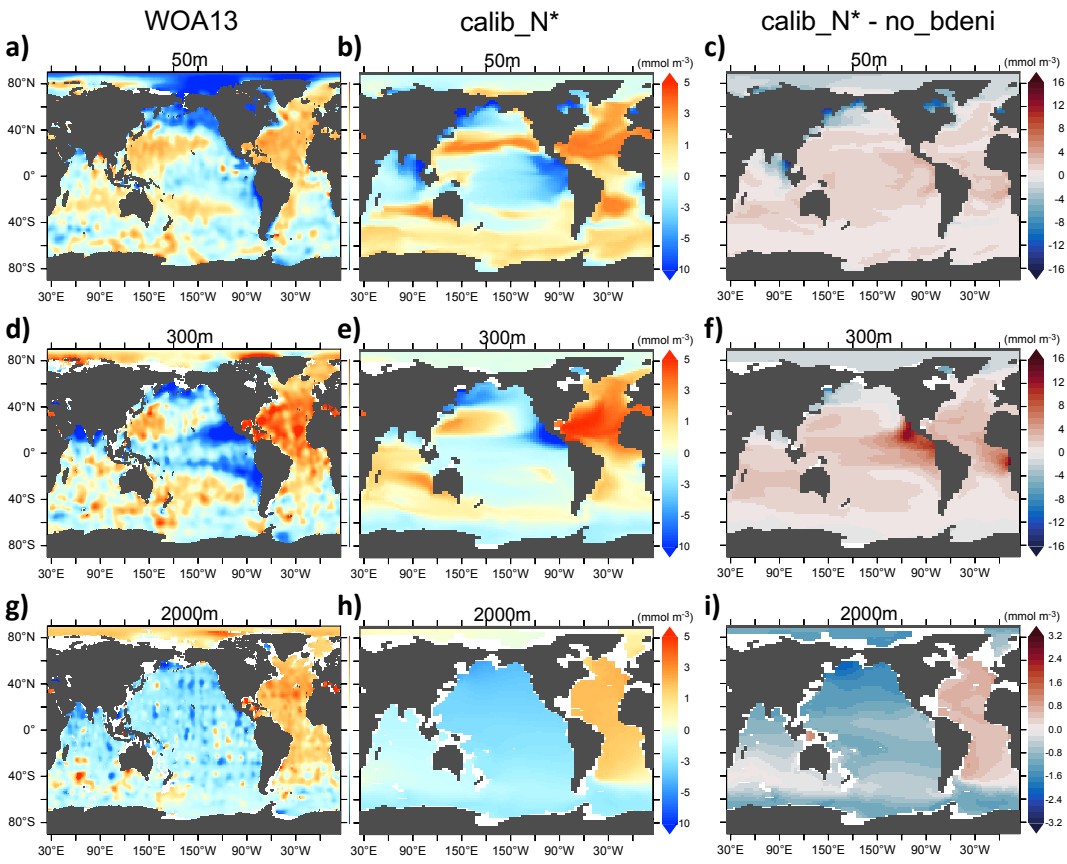

**Figure 7.** Spatial distribution of N* (50m, 300m, 2000m) for WOA13 and one of our calibrated solutions calib_N*, as well as changes in this calibrated solution compared to no_bdeni.

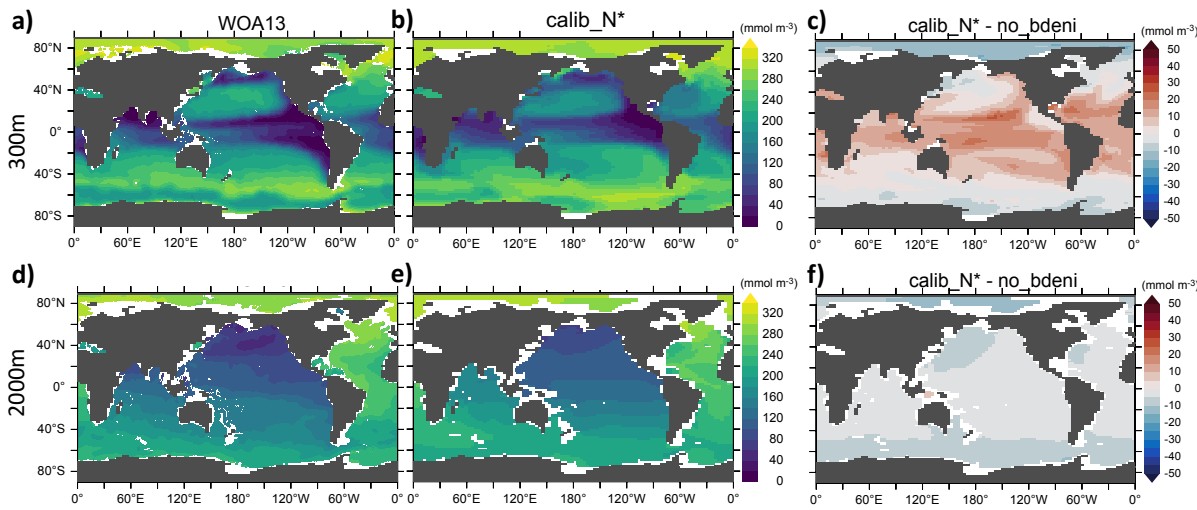

**Figure 8.** Spatial distribution of oxygen concentration at 300m and 2000m for calib_N* and the changes of this new solution relative to no_bdeni.

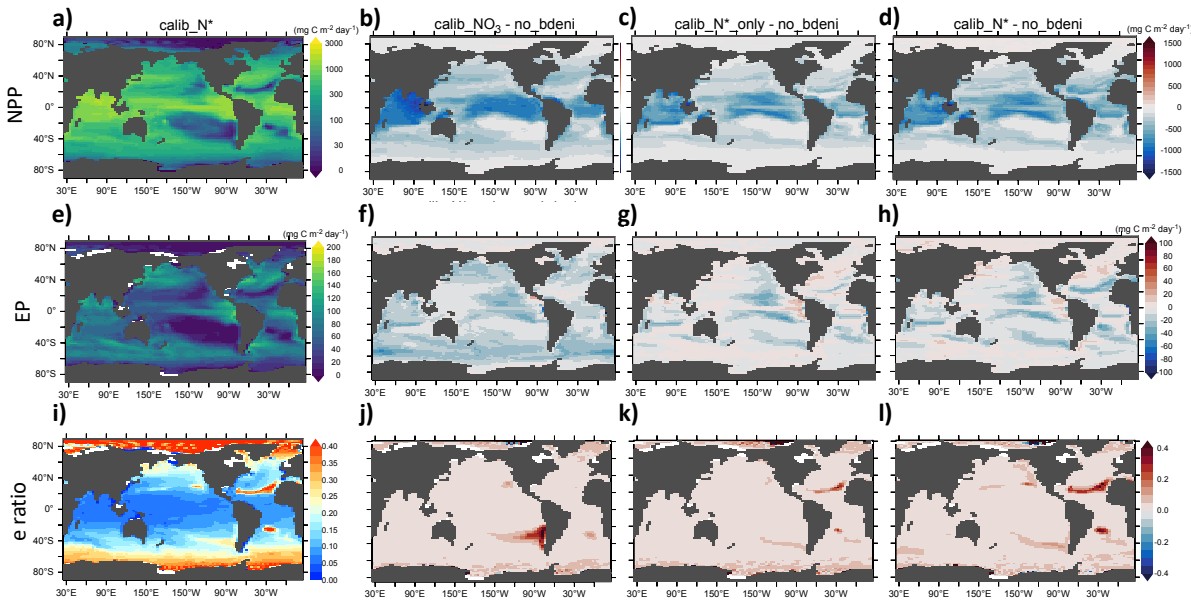

**Figure 9.** Vertically-integrated net primary production (NPP), export production (EP) at 130m, and e-ratio distributions for calib_N* and the changes of three new solutions relative to no_bdeni.

*Code and data availability.*   The University of Victoria Earth System Climate Model version 2.9 is available at https://terra.seos.uvic.ca/model/ (last access: 22 November 2023) ("updated model" version). The benthic-denitrification-included model and OPEM codes, Matlab scripts of cost function, and data shown in the figures and used for this paper are available at https://doi.org/10.5281/zenodo.10469908.

**Appendix A**

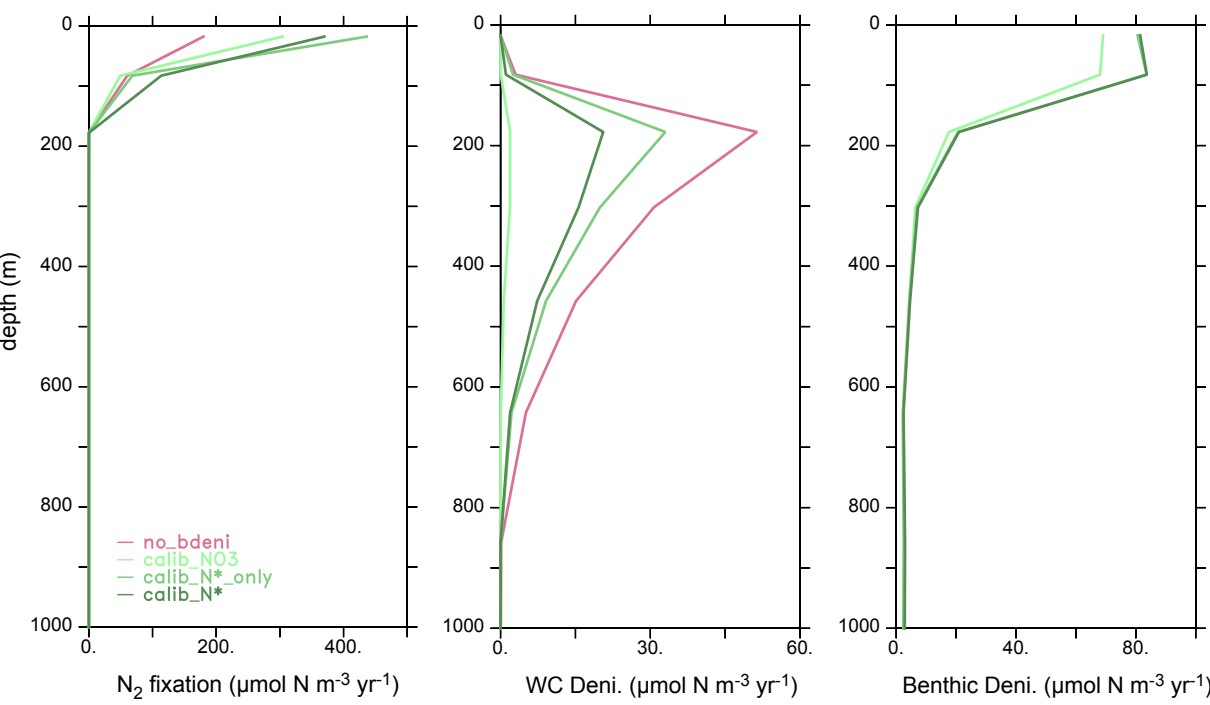

**Figure A1.** Vertical profiles of the global averages of major N fluxes at upper 1000m.

**Table A1.** vertically integrated N fluxes, excess N relative to P in the organic matter (PON*), and average total excess N relative to P including organic and inorganic forms ($\overline{TN*}$) for ocean basins.

|  | Atlantic | Pacific | Indian | SO |
|---|---|---|---|---|
|  | | d2_6 | | |
| N$_2$ Fixation | 43.2 | 21.1 | 24.3 | 0 |
| WC Deni | 0 | 2.9 | 0 | 0 |
| B Deni | 25.4 | 27.6 | 22 | 13.0 |
| Net | 17.7 | –9.4 | 2.3 | –13 |
| PON* | 309.3 | 12.3 | 16.1 | 11.9 |
| $\overline{TN*}$ | –1.1 | –3.3 | –2.5 | –2.6 |
|  | | f2_12 | | |
| N$_2$ Fixation | 56.4 | 46.9 | 31.9 | 0 |
| WC Deni | 6.6 | 35.3 | 0.2 | 0 |
| B Deni | 28.5 | 26.9 | 22.7 | 12.6 |
| Net | 21.3 | –15.3 | 9.0 | –12.6 |
| PON* | 522.9 | –151.8 | 69.7 | –21.2 |
| $\overline{TN*}$ | 0.5 | –2.2 | –1.0 | –1.2 |
|  | | b2_4 | | |
| N$_2$ Fixation | 53.5 | 48.4 | 34.2 | 0 |
| WC Deni | 3.3 | 26.8 | 0 | 0 |
| B Deni | 30 | 31.4 | 23.4 | 16.5 |
| Net | 20.2 | –9.8 | 10.8 | –16.5 |
| PON* | 548.5 | –18.7 | 147.4 | –279.6 |
| $\overline{TN*}$ | 0.7 | –2.6 | –1.1 | –1.5 |

|  | Atlantic | Pacific | Indian | SO |
|---|---|---|---|---|
|  | | no_bdeni | | |
| N$_2$ Fixation | 27.1 | 31.1 | 11.0 | 0 |
| WC Deni | 11.3 | 56.1 | 0 | 0 |
| Net | 15.7 | –25.0 | 10.9 | 0 |
| PON* | 104.4 | –416.1 | 53.2 | –269.3 |
| $\overline{TN*}$ | 0.4 | –2.0 | –0.8 | –1.0 |

where PON* = PON – 16*POP and TN* = (PON + $NO_3^-$) – 16 * (POP + $PO_4^{3-}$) + 2.9 = N* + PON*.
The unit for fluxes is $\mathrm{TgNyr}^{-1}$ and that of the vertically-integrated PON* is $10^{12}$ mmol m$^{-3}$.

*Author contributions.* C.J.S., A.L. and M.P. designed the study. N.L., C.C. and M.P. carried out the simulations and calibration. N.L. conducted the analysis. All authors discussed the results and wrote the manuscript.

*Competing interests.* The authors declare no competing financial interest.

*Acknowledgements.* We are grateful to M.S. for sharing the code of the original cost function. N.L. was supported by the the China Scholarship Council (CSC; grant no. 201906330071). C.J.S. was supported by the Deutsche Forschungsgemeinschaft (DFG project number 445549720). C.C. was supported by the DFG project 447794210.

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
