# Peer review of "Global impact of benthic denitrification on marine $N_2$ fixation and primary production simulated by a variable-stoichiometry Earth system model"

_EGUsphere, 2024_

## Author Comment (AC1)

**Response to egusphere-2024-123 RC1**

Below is our response to RC1. Referee #1′s comments are in **bold**. Our responses are in black. Changes to the manuscript are indicated in blue. The line numbers refer to the previously submitted manuscript without track changes.

**In this manuscript, Li et al have implemented benthic denitrification using the empirical function derived by Bohlen et al (2012) on Uni-Vic ESM, and parameterized using chl, O2, NO3, PO4, N\* observations. Relatively speaking they found considering N\* to constrain their model has given better model performance when compared to the observations from WOA13. They found that this implementation has resulted in better simulation of N2 fixation patterns in the world ocean, and more realistic simulation of ODZs, suggesting that inclusion of benthic denitrification in ESMs are a key piece to improve model performance.**

**I found the procedure and inferences are sound. This work also highlights the importance of consideration of benthic processes in ESMs which are currently underrepresented, and its potential in improving model predictions. I have no major concerns with the present study; however, I feel the following comments will increase the readability of the paper.**

We thank the reviewer for these very positive and helpful comments.

**Minor comments:**

**Line 23: replace "however, even though" with in spite of.**

[revised manuscript text omitted]

NEW: "The global oceanic fixed-N inventory is maintained by the balance of $N_2$ fixation in the surface ocean and water-column and benthic denitrification. "

**Line 279-280: Not clear to me. Please explain. In figure 8, Calib_N* - no_bdeni values are mostly positive, that means Calib_N* has more O2 in the ocean, I see that in figure 6 as well. But how does the prediction of ODZs in no_bden case. If is a bit confusing, please clarify in the text.**

To make it clear, we will add a row of ODZ volumes of all simulations in Table 2. If we define ODZ as the region with $O_2$ concentration < 5 mmol m$^{-3}$, the ODZ volume in no_bdeni is $>10^{15}$m$^3$ , whereas that of calib_N* is $6.58*10^{14}$m$^3$.

OLD: "This implies that including benthic denitrification may improve the representation of ODZs in global ocean biogeochemical models that typically overestimate their volume (Cabré et al., 2015)."

NEW: "The lower ODZ volume when including benthic denitrification (Table 2) implies that including benthic denitrification may improve the representation of ODZs in global ocean biogeochemical models that typically overestimate their volume (Cabré et al., 2015)."

**287-298: It will be nice to add a third row of figures on figure 9 plotting the EP ratio. Also, can you comment on the EP ratio how it compares to any other literature (observation or modeling)? It is good to put this information as the rain rate to the seafloor will ultimately drive the denitrification in the model.**

Thank you for the suggestion. We will add the e-ratios in Figure 9.

[Figure]

We will add the description of spatial distribution of e-ratio and comparison with other literature in L291: "The global distribution of export efficiency (EP/NPP, hereafter e-ratio) exhibits a negative relationship between NPP and e-ratio. Low e-ratios (<0.2) in low latitudes and high e-ratios (>0.2) in high latitudes, with a few exceptions in the Atlantic and Pacific subtropical gyres (Fig. 9i). Its general pattern is similar to the observational estimate of Dunne et al. (2005) and the model estimate of Henson et al. (2015). "

**305: I think you have not put any number of observational estimates of N fixation, if not please provide some number in relevant places.**

OLD: "The absence of a persistent ODZ in the Arabian in our model is likely a reason why our global water-column denitrification and $N_2$ fixation rates are on the low-end of observational estimates."

NEW: "The absence of a persistent ODZ in the Arabian Sea in our model is likely a reason why our global water-column denitrification and $N_2$ fixation rates are on the low-end of observational estimates range from 131 to 253 Tg N yr−1 (Großkopf et al., 2012; Luo et al., 2012; Landolfi et al., 2018; Shao et al., 2023)."

To make it easier to compare our estimates with other observational estimates, we now indicate the observational estimates with an uncertainty range of 131 to 253 Tg N yr−1  as a shaded grey area in Fig.2.

**Define what is biome when it first occurs.**
Line 119: We will add "Ocean biomes are geographical regions characterized by coherent large-scale patterns in physical and biogeochemical functions (Fay and McKinley, 2014), providing a representation of global ocean biogeography."

**If the Arabian sea has one, how does it compare with other modelled areas/seas?**

Thank you for the question. In the cost function, each biome is characterised by two numbers for each depth layer (and month for the upper 5 layers of UVic): the spatial average and variance within the biome.In this study, we consider the Indian Ocean as a single biome including the Arabian Sea, thus the spatial average and variance of physical and

biogeochemical variables within the whole Indian Ocean is taken into account in the cost function. If we consider the Arabian Sea to be one biome, the model calibration will explicitly reflect the spatial average and variance of its physical and biogeochemical variables, providing the same importance in cost function as other biomes.

We will clarify further the calculation of the cost function in Section 2.4.2 (Line 116) as below.

OLD: "To assess the model performance with respect to the spatial distributions of dissolved tracers and surface chlorophyll a, we apply global misfit metrics J based on a maximum-likelihood estimation (ML) method for parameters, assuming log-normally distributed errors, with observations grouped by distinct biogeochemical biomes (Fay and McKinley, 2014), as described in Chien et al. (2020). Briefly, the calculation of our cost function comprises two terms for every depth level in our model,

$$J_k = d_k^T R_k^{-1} d_k + v_k^T V_k^{-1} v_k, k \in \{1, 2, \ldots, 19\}$$

(1)

where the residual (d) between observations (o) and the model (m) within each biome is defined as d = o − m, and the discrepancy in the spatial variance (v) as v = v(o) − v(m). For each variable included in the cost function, we normalize the concentrations to threshold values and log-transform to achieve approximately Gaussian error distributions for d and v with zero mean and covariance matrices R and V, respectively. We integrate the results across the 17 biomes, applying Eq. (1) to monthly means for the upper 550m (top 5 layers in UVic) and annual means below (bottom 14 layers)."

NEW: "To assess the model performance with respect to the spatial distributions of dissolved tracers and surface chlorophyll a, we apply global misfit metrics *J* based on a maximum-likelihood estimation (ML) method for parameters, assuming that the errors for the residuals of log-transformed variables between model simulations and observations follow normal distributions (Chien et al., 2020). Minimizing *J* ensures the best parameter estimates for the given model configuration.

Instead of calculating residuals between model simulations and observations for each model grid cell, we categorize model simulations and observations into 17 biomes (Fay and McKinley 2014), . Ocean biomes are geographical regions characterized by coherent large-scale patterns in physical and biogeochemical functions (Fay and McKinley 2014), providing a representation of global ocean biogeography. We include two statistical measures of variables at each biome to represent it: spatial average and variance. Therefore, the residuals consist of the discrepancies in both spatial average and variance of each biome. In the vertical spatial dimension, we do not make any simplifications and residuals are calculated at each depth layer (k). We calculate the residuals of variables between monthly-averaged simulations and observations to resolve seasonal variations in the upper ocean (0-550m), and between annually-averaged simulations and observations below 550m.

Thus, the calculation of our cost function comprises two components,

$$J_k = A_k^T \Sigma_k^{-1} A_k + V_k^T Q_k^{-1} V_k, \quad k \in \{1, 2, \ldots, 19\}$$

(1)

where *A* is the residual between the spatial means of observations (*o*) and the model (*m*) defined as *A* = mean(*o*) − mean(*m*) and *V* is the residual in the spatial variance (var) as *V* = var(*o*) − var(*m*). The covariance matrices of *A* and *V* are denoted by **Σ** and **Q**, respectively."

**325-326: Define what is $w_{d0}$ and $w_{dd}$, when it appears first.**

$W_{dd}$ is defined at Line 96 and we will modify Line 325–326 as below.

OLD: "Improvements to the upper ocean may also have the potential to improve the deep-ocean performance, such as including $w_{d0}$ into the calibrated parameters. By taking both $w_{d0}$ (particle sinking velocity at the surface ocean) and $w_{dd}$ into account for the calibration, …"

NEW: "Improvements to the upper ocean may also have the potential to improve the deep-ocean performance, such as including the sinking speed of particles at the ocean surface ($w_{d0}$) in addition to its increase with depth ($w_{dd}$) into the set of calibrated parameters. By taking both $w_{d0}$ and $w_{dd}$ into account for the calibration, …"

**339-340: first part of this sentence is not clear.**

OLD: "In order to explore the sensitivity to prior assumptions, we applied our cost function to calibrate model solutions against three different combinations of observations."

NEW: "In order to explore the sensitivity to prior assumptions about which data are used for calibration, we applied our cost function to three different combinations of observations."

**345: "demonstrates"**

OLD: "The greater constraining capacity of N* in comparison to considering nitrate and phosphate separately highlights the importance of accounting for correlations among variables within the cost function (Krishna et al., 2019) and demonstrations the power of diagnostic tracers such as N* for diagnostic studies of the ocean nitrogen cycle (DeVries et al., 2013; Eugster and Gruber, 2012; Deutsch et al., 2007)."

NEW: "The greater constraining capacity of N* in comparison to considering nitrate and phosphate separately highlights the importance of accounting for correlations among variables within the cost function (Krishna et al., 2019) and demonstrates the power of diagnostic tracers such as N* for diagnostic studies of the ocean nitrogen cycle (DeVries et al., 2013; Eugster and Gruber, 2012; Deutsch et al., 2007)."

**Figure 2: Any reference for NPP to compare with?**

Figure 2: we will mark published estimates of NPP as grey area.

**Figure 5: explain why you did not include Indian ocean and southern ocean.**

Figure 5: we will add the Indian and Southern Ocean.

Section 3.3.2: add description of Indian and Southern Ocean.

**Figure 6 caption: Typo.**

OLD: "Vertical distributions of tracers ($NO_3^-$, $PO_3^{4-}$, N*, $O_2$) in global and respect basins (Atlantic, Pacific, Indian, Southern Oceans)."

NEW: "Vertical distributions of tracers ($NO_3^-$, $PO_3^{4-}$, N*, $O_2$) in the global ocean and the Atlantic, Pacific, Indian, Southern Oceans."

**Figure 8 caption: three solutions or N*?**

yes, it should be solution calib_N*. We removed "of three new solutions".

**Table 1 caption: units. Typo.**

Table 1 caption: Parameter names, ranges, units and descriptions.

**Table 1: did you get the reference range from somewhere? please clarify.**

yes, the reference range comes from Pahlow et al., 2020 and Chien et al., 2020. We added the clarification "The reference range refers to the calibration range in Pahlow et al. (2020); Chien et al. (2020)" below the table.

**Table A1: Why do you provide bar on top of TN? It is not defined in the table.**

The bar on top of TN* means the average of TN* of each basin. We added the definition in the table title.

---

## Author Comment (AC2)

**Response to egusphere-2024-123 RC2**

Below is our response to RC2. Referee #2′s comments are in **bold**. Our responses are in black. Changes to the manuscript are indicated in blue. The line numbers refer to the previously submitted manuscript without track changes.

**This manuscript describes the improvement of an Earth system model by incorporating benthic denitrification, highlighting the importance of benthic denitrification in shaping the global distributions of NPP, N2 fixation, oxygen, etc. In addition, the authors conducted a large ensemble of simulations and applied the Latin-hypercube sampling method to choose the model parameters. The simulation identified that the N* is an essential parameter to calibrate the model, which has an important implication for future model calibration. Overall, the manuscript was well-written and organized, and the references cited are up-to-date and appropriate. I recommend publication after making the following clarifications and modifications.**

We thank Referee #2 for the positive feedback.

**I'd like to see more discussion on the pattern of N2 fixation. Currently, the distribution patterns in the S. Pacific and N. Indian Ocean (Fig. 4a) seem not correct to my eyes. What causes such distribution, is there anything to do with benthic denitrification?**

We agree that our model may underestimate $N_2$ fixation rates in the South Pacific and N. Indian Ocean (Fig. 4a). However, it is unlikely that the underestimation stems from benthic denitrification, since the differences between new simulations with no_bdeni (Fig. 4b-d) indicate there is no strong pattern shift in these basins compared to the changes in other subtropical regions. We suspect there are two different dominant processes that contribute to the underestimation of the $N_2$ fixation rates at these two basins separately.

In our model, diazotroph growth and $N_2$ fixation in the South Pacific are subjected to the low Fe availability indicated by the low dissolved Fe concentration (see the left panel of Figure below), especially at the South Pacific Subtropical Gyre. Our simulated surface Fe concentration in the South Pacific Subtropical Gyre is lowest among all subtropics, whereas observations and other models (Huang et al., 2022) suggest that the surface dissolved Fe in the South Pacific is comparable to that in the North Pacific. We further assess the effect of benthic denitrification on dissolved Fe concentration (see the right panel of Figure below). In the South Pacific, the differences of surface dissolved Fe concentrations between calib_N* and no_bdeni shows only minor changes. This minimal influence of benthic denitrification on dissolved Fe suggests that benthic denitrification does not affect $N_2$ fixation through Fe concentration in this region.

The absence of nitrogen fixation in the Northern Indian Ocean may result from the absence of an oxygen-deficient zone (ODZ) and water-column denitrification in this area in our model. A proper representation of the ODZ in the Arabian Sea is one of our goals of ongoing model development.

We will add on Line 213: "The very low $N_2$ fixation rates in the South Pacific (Fig. 4a) can be attributed to the underestimated surface dissolved iron concentration in this region."

We will add on line 207: "This is also the main reason for the very low rates of $N_2$ fixation predicted for the Northern Indian Ocean, which could be unrealistic. Yet, there is considerable uncertainty about the regional pattern of $N_2$ fixation in the Northern Indian Ocean based on the

sparsity of available observations. For example, Shao et al. (2020) found strong N$_2$ fixation rates at only a few of places along the southwest coast of India in the eastern Arabian Sea, Löscher et al. (2020) could find no evidence for N$_2$ fixation in the Bay of Bengal, and vast areas in the Northern and Western Indian ocean remain unsampled."

[Figure]

**In section 4 "Model and calibration limitations", the authors discussed multiple ways to improve the model, including applying different parameterization schemes for upper and deeper ocean sinking speed and resolving the dependency of remineralization rate on O2. The methods described are all reasonable. But why not try them, this is part of model development.**

This research concentrates on the impact of benthic denitrification on global ocean biogeochemistry in our model. A full investigation of the possible development discussed in Section 4 requires calibrating the model configurations for each change, which is very time-consuming and goes beyond the scope of this work. We plan to do some of these in the future.

**Some minor suggestions**

**Line 83 (Somes and Oschlies, 2015) --> Somes and Oschlies, (2015)**

ok

**Line 92 (Pahlow et al., 2020; Chien et al., 2020) --> Pahlow et al., (2020) and Chien et al., (2020)**

ok

**Line 149 Best models choices --> Best model choice.**

ok

**Line 305 Arabian --> Arabian Sea**

ok

References

Huang, Yibin, Alessandro Tagliabue, and Nicolas Cassar. 2022. "Data-Driven Modeling of Dissolved Iron in the Global Ocean." *Frontiers in Marine Science* 9. https://doi.org/10.3389/fmars.2022.837183.

Löscher, Carolin R., Wiebke Mohr, Hermann W. Bange, and Donald E. Canfield. 2020. "No Nitrogen Fixation in the Bay of Bengal?" *Biogeosciences* 17 (4): 851–64.

Shao, Zhibo, Yangchun Xu, Hua Wang, Weicheng Luo, Lice Wang, Yuhong Huang, Nona Sheila R. Agawin, et al. 2023. "Global Oceanic Diazotroph Database Version 2 and Elevated Estimate of Global Oceanic N2 Fixation." *Earth System Science Data* 15 (8): 3673–3709.